# Identification of Genomic Regions Conferring Enhanced Zn and Fe Concentration in Wheat Varieties and Introgression Lines Derived from Wild Relatives

**DOI:** 10.3390/ijms251910556

**Published:** 2024-09-30

**Authors:** Irina N. Leonova, Antonina A. Kiseleva, Elena A. Salina

**Affiliations:** The Federal Research Center, Institute of Cytology and Genetics SB RAS, Novosibirsk 630090, Russia; antkiseleva@bionet.nsc.ru (A.A.K.); salina@bionet.nsc.ru (E.A.S.)

**Keywords:** wheat, zinc, iron, GWAS, wheat relatives, *Triticum timopheevii*

## Abstract

Wild and cultivated relatives of wheat are an important source of genetic factors for improving the mineral composition of wheat. In this work, a wheat panel consisting of modern bread wheat varieties, landraces, and introgression lines with genetic material of the wheat species *Triticum timopheevii*, *T. durum*, *T. dicoccum*, and *T. dicoccoides* and the synthetic line *T. kiharae* was used to identify loci associated with the grain zinc (GZnC) and iron (GFeC) content. Using a BLINK model, we identified 31 and 73 marker–trait associations (MTAs) for GZnC and GFeC, respectively, of which 19 were novel. Twelve MTAs distributed on chromosomes 1B, 2A, 2B, 5A, and 5B were significantly associated with GZnC, five MTAs on 2A, 2B, and 5D chromosomes were significantly associated with GFeC, and two SNPs located on 2A and 2B were related to the grain concentration of both trace elements. Meanwhile, most of these MTAs were inherited from A^t^ and G genomes of *T. timopheevii* and *T. kiharae* and positively affected GZnC and GFeC. Eight genes related to iron or zinc transporters, representing diverse gene families, were proposed as the best candidates. Our findings provide an understanding of the genetic basis of grain Zn and Fe accumulation in species of the Timopheevi group and could help in selecting new genotypes containing valuable loci.

## 1. Introduction

Mineral malnutrition is a significant issue in many countries around the world, regardless of their level of economic development. Currently, more than two billion people globally have inadequate intakes of macro- and micronutrients and vitamins necessary for a healthy lifestyle [1]. For the proper functioning of all human body systems, at least nine essential microelements are required: iron, zinc, selenium, copper, manganese, molybdenum, iodine, chromium, and cobalt, with iron (Fe) and zinc (Zn) being the most important [2,3].

Iron is one of the vital microelements, serving as a cofactor in various enzymatic reactions, participating in energy transfer, and playing a crucial role in the formation of hemoglobin for oxygen transport [3]. Iron deficiency in the human body is one of the primary causes of anemia. The World Health Organization estimates that in 2019, 571 million of women aged 15–49 years, 32 million pregnant women, and 269 million children aged 6–59 months were affected by anemia, with the African continent and the Southeast Asia region being the most impacted [4,5]. Iron deficiency anemia affects more than 50% of pregnant women and children living in South and Central America [6].

Zinc is the second most essential trace element, necessary for the proper functioning of various biochemical reactions and transcription processes. It plays a role in cell proliferation, regulates insulin activity, and, consequently, affects blood glucose levels [7,8]. Zinc deficiency can lead to stunted human growth and development, reproductive organ dysfunction, autoimmune and allergic diseases, reduced visual acuity, and other health issues [9].

Bread wheat (*Triticum aestivum* L.) is the primary source of protein and mineral elements for the majority of the population worldwide. By consuming wheat products, people receive, on average, up to 20–30% of their daily calories, up to 15% of their daily iron, and up to 11% of their daily zinc [10,11,12,13]. However, the increase in gross grain harvest due to the introduction of high-yielding wheat varieties has been accompanied by a decrease in the protein and gluten contents, as well as the concentration of micro- and macroelements, which provide the nutritional value of the final product [14].

An extensive reservoir of genetic diversity for grain iron (GFeC) and zinc (GZnC) concentration is found in the cultivated and wild relatives of bread wheat. The Fe and Zn contents in the grain of modern wheat varieties vary considerably but are generally lower than those in related species [15,16,17,18,19]. It has been established that many accessions of the genus *Aegilops* (*Aegilops searsii*, *Ae. umbellulata*, *Ae. caudata*, *Ae. geniculata*, etc.) contain 2–3 times higher concentrations of Zn and Fe compared to modern wheat cultivars [20,21]. A wide range of variations in GFeC and GZnC is characteristic of wild emmer (*T. dicoccoides*), among which some varieties have been found to combine high concentrations of zinc, iron, and protein in the grain [22,23]. Increased levels of Fe and Zn have also been detected among accessions of the diploid species *T. monococcum* and *T. boeoticum*, which were used to create mapping populations and subsequently identify of loci on chromosomes 2A (Fe + Zn) and 7A (Fe) [24].

The introgression of beneficial alleles from wild relatives requires a considerable amount of time to eliminate undesirable alien genetic material. Introgression lines, along with additional and substituted lines, provide unique plant material for the identification of effective genes/QTLs. These lines, containing valuable genes, can be considered as a “bridge” for the subsequent transfer of target loci. Synthetic hexaploid wheat lines (SHWs), derived from the hybridization of various accessions of durum wheat (*T. turgidum* ssp. *durum*) with wild goat grass (*Ae. tauschii*), are considered potential sources of genes/QTLs for improving the mineral composition of bread wheat grain. Several studies indicate that the concentration of Fe and Zn in the grain of individual SHWs is 20–30% higher than that in bread wheat [19,25,26]. The study of introgression, substitution, and additional lines obtained with the participation of *Aegilops* ssp., *T militinae*, *Thinopyrum bessarabicum*, and *T. kiharae* identified forms with a high content of minerals, including zinc and iron [27,28,29,30]. Polish wheat (*T. polonicum*) has been proposed as promising material for the biofortification of bread wheat, as most accessions of this species have been characterized by the highest content of copper, iron, and zinc and the lowest concentrations of strontium, aluminum, and barium in comparison with *T. aestivum* and *T. durum* [31,32].

To date, in the germplasm of bread wheat, using biparental mapping populations and GWAS, the genetic control of GFeC and GZnC has been studied, the chromosomal localization of loci has been determined, and molecular markers linked to QTLs and recommended for marker-assisted selection have been identified [33,34,35,36,37]. A comprehensive study to identify key genomic regions for Zn and Fe biofortification was conducted by Juliana et al. [38], who, using a panel of 5585 advanced bread wheat pre-breeding lines, identified 141 markers on all wheat chromosomes except chromosomes 3A and 7D. New QTLs for GFeC and GZnC were mapped using SHW collections based on *T. durum* and *T. dicoccum* [39,40], mapping populations obtained from crossing durum wheat with wild emmer [41], as well as wheat–*Aegilops* substitution and addition lines [42].

Currently, there is very little information on the grain mineral composition of species in the Timopheevi lineage with A^t^A^t^GG genomes (*T. timopheevii*, *T. araraticum*, and *T. zhukovskyi*). The few available results indicate that some accessions of these species exhibit favorable characteristics in terms of grain mineral quality and could be used in the biofortification of bread wheat [43,44]. We have previously shown that recombinant lines of bread wheat obtained with the participation of *T. timopheevii* contain new loci associated with increased gluten content in grain [45]. The present study aimed to analyze GFeC and GZnC in a wheat collection consisting of bread wheat varieties and introgression lines with genetic material transferred from wheat relatives and to identify genomic regions associated with these traits.

## 2. Results

### 2.1. Phenotypic Variation in GFeC and GZnC

The variability of Fe and Zn content in wheat grain was observed over two consecutive years (2018–2019) using a panel of wheat varieties and introgression lines obtained from crossing bread wheat with *T. timopheevii*, *T. durum*, *T. dicoccum*, *T. dicoccoides*, and *T. kiharae* (Appendix A). It was shown that the iron content in wheat grain ranged from 28.2 mg/kg to 70.2 mg/kg in 2018 and from 26.9 mg/kg to 75.7 mg/kg in 2019 (Figure 1). Zinc content exhibited greater variability, ranging from 22.6 mg/kg to 63.5 mg/kg in 2018 and from 27 mg/kg to 100.8 mg/kg in 2019 (Appendix A). For both traits, a reliable increase in mean values (*p* < 0.000001) was observed in 2019 compared to 2018. The broad-sense heritability of the traits was high in both years, varying from 0.67 for Zn in 2019 to 0.89 for Fe in 2018 (Appendix A). A strong positive significant correlation was found between Fe and Zn content in 2018 (r = 0.56, *p* < 0.001) and 2019 (r = 0.57, *p* < 0.001). Both traits were significantly (*p* < 0.001) influenced by genotype (G), environmental conditions (E), and the GxE interaction (Appendix A).

Comparative analysis of GFeC and GZnC in wheat cultivars, introgression lines (ILs), and wheat relatives (WRs) indicated that the concentration of these trace elements was significantly higher in ILs than in wheat cultivars (Figure 2). Among the bread wheat varieties, genotypes Provintsiya, Novosibirskaya-22, Aleshina, Belorusskaya-80, and Kinelskaya-60 exhibited relatively high GFeC and GZnC (>50 mg/kg), with the cultivars Aleshina and Provintsiya showing notably high levels of both Fe and Zn. Higher Zn concentrations of (50–70 mg/kg) were observed in the grain of ILs containing introgressions from *T. timopheevii*, *T. kiharae*, and *T. dicoccum*. The highest GFeC, ranging from 50 to 67 mg/kg, was found in ILs derived from crosses with *T. timopheevii* and *T. kiharae*. Spearman’s correlation analysis revealed a high level of relationship between the GFeC and GZnC based on BLUEs (r = 0.57, *p* < 0.001).

### 2.2. Results of GWAS

Summarizing the results of all the tested models, thirty-one SNPs on chromosomes 1A, 2A 2B, 3B, 5A, 5B, 5D, and 7A were significantly (*p* < 0.001) associated with GFeC. Seventy-three SNPs were found to be significantly (*p* < 0.001) associated with GZnC content located on chromosomes 1A, 1B, 1D, 2A, 2B, 4A, 4B, 5A, 5B, 6A, 6D, 7A, 7B, and 7D (Figure 3, Appendix A).

Based on the results of the LD analysis between significant markers (Appendix A), twenty loci were identified on chromosomes 1A, 1B, 1D, 2A, 2B, 3B, 4A, 5B, 6A, 7A, and 7B, containing from two to eleven SNPs associated with GFeC or GZnC. Two of these loci (*QZnFe.icg-2A* and *QZnFe.icg-2B*) were associated with the content of both trace elements (Appendix A).

### 2.3. Significant MTAs Inherited from Wild Wheat Relatives

Analysis of the allelic variation of SNPs associated with GFeC and GZnC allowed us to identify haplotypes inherited from *T. timopheevii*, *T. kiharae*, or *T. dicoccum*, which are predominantly present in the genome of ILs. In contrast, the frequency of wild haplotypes in the bread wheat cultivars from this collection was less than 5%. In total, 19 such SNPs were detected, with 12 associated with GZnC, 5 with GFeC, and 2 (*Kukri_c441_891* and *BobWhite_c5191_563*) included in the *QZnFe.icg-2A* and *QZnFe.icg-2B* loci being associated with the grain concentration of both trace elements (Table 1 and Appendix A).

According to LD data, we composed four loci on chromosomes 1B (*QZn.icg-1B.1*, *QZn.icg-1B.2*), 2A (*QZn.icg-2A*), and 2B (*QZnFe.icg-2B*), consisting of two to four linked SNPs (Table 1, Appendix A). Five SNP markers located within a narrow interval of 56.64–57.59 cM, according to the genetic consensus map of wheat, formed two loci on chromosome 1B (*QZn.icg-1B.1* and *QZn.icg-1B.2*). The presence of the GTA haplotype (locus *QZn.icg-1B.1*), formed by the markers *BS00025736_51*, *Kukri_rep_c100936_449*, and *GENE-0427* and inherited from *T. timopheevii*, *T. kiharae*, and *T. dicoccum*, had a positive effect on GZnC. Conversely, the alleles of the markers *Excalibur_c17202_1833* and *RAC875_c826_839*, forming the *QZn.icg-1B.2* locus and originating from *T. timopheevii* and *T. kiharae* (haplotype GG), had a negative effect on zinc concentration (Table 1).

Three tightly linked SNPs (*Kukri_c44442_274*, *Kukri_c441_891*, and *TA001792-1026*) significantly associated with the analyzed traits were found on the short arm of chromosome 2A, forming the locus *QZnFe.icg-2A* within the region 123.5–141.1 Mb. The positions of these SNPs on the genetic linkage map of chromosome 2A ranged from 102.43 to 103.62 cM. These three markers formed the haplotype CGG, which is specific to *T. timopheevii*, *T. kiharae* and 19 ILs derived from their crosses, and had a positive effect on Fe and Zn concentrations (Table 1).

Locus *QZnFe.icg-2B* included four markers on chromosome 2B within the region from 767.13 Mb to 767.37 Mb, with *BobWhite_c5191_362*, *BobWhite_c5191_563*, *CAP12_s9114_61*, and *BS00046601_51* located close to each other. On the consensus genetic linkage map, these SNPs were colocalized at the same position, 134.45 cM. The haplotype “TTTC”, identified in *T. timopheevii*, *T. kiharae*, and 12 introgression lines, was associated with increased Fe content compared to the *T. aestivum* haplotype “CCCT” (Table 1).

Except for the loci formed by linked SNPs, there were seven single SNPs on chromosomes 1B, 2B, 5A, 5B, and 5D, which were likely inherited from the WRs and associated with the content of the studied trace elements (Table 1). Alleles of SNPs characteristic of all three WRs in this study (*T. timopheevii*, *T. kiharae*, and *T. dicoccum*), such as *Ku_c6546_718*, *BS00067150_51*, and *BS00091302_51*, were associated with higher Zn content, while *RAC875_c703_2551* was associated with higher Fe content. Additionally, *Excalibur_c14859_394*, found in *T. timopheevii* and *T. kiharae*, was also associated with higher GZnC. In contrast, the alleles *Tdurum_contig64416_441* and *Excalibur_c82693_359* of these two species were related to a decrease in GZnC.

### 2.4. Candidate Genes

Gene prioritization was performed for QTLs defined by markers selected with a significance threshold of *p* < 0.001. In total, there were 2002 genes associated with these QTLs according to the IWGSC RefSeq v2.1 (Appendix A). The number of candidate genes was reduced to 111 after Gene Ontology analysis. Further analysis of traits associated with these genes using KnetMiner [48] and a search for members of protein families known to be involved in iron and zinc content revealed eight genes, which we propose as the best candidates involved in change of zinc and iron concentration in wheat grain (Table 2). The expression patterns of these selected prioritized genes in different tissues during wheat plant development are presented in Appendix A.

## 3. Discussion

### 3.1. Comparison of Putative QTL Localization with Known Data

The genome-wide association study (GWAS) conducted in this research aimed to uncover novel genetic loci potentially linked to zinc and iron content in wheat grain. Previously, numerous markers and loci associated with iron and zinc content in wheat grain were identified across all 21 wheat chromosomes [35,36,49,50,51,52]. According to the results of the two-year field evaluation, we also identified associations with GZnC and GFeC distributed across chromosomes 1A, 1B, 1D, 2A, 2B, 3B, 4A, 5A, 5B, 5D, 6A, 6D, 7A, and 7B. Notably, the most significant effects on the traits were observed for associations on chromosomes 1A, 1B, 2A, 2B, 3B, 4A, 5A, and 5B (Appendix A).

We specifically focused on peak SNPs to determine their overlap with markers identified in our study. Wang et al. [53] identified 911 SNPs significantly associated with grain Fe concentrations. The authors, analyzing a collection of wheat varieties of different geographical origin, revealed two SNPs (*AX-108896742* and *AX-95247517*) for GFeC located in the region of 105 Mb and 182 Mb of chromosome 2A, respectively; however, their localization did not coincide with *QZnFe.icg-2A* as mapped in our work. Potapova et al. [54] detected four highly significant SNPs associated with micro- and macroelement contents in wheat grain through multivariate and meta-analyses. A marker on chromosome 6A associated with the content of iron and manganese was found to be close to the SNP marker *GENE-4178_89* discovered in our study (Appendix A). Locus *QFe.icg-5B.2* (Appendix A), responsible for Fe content and located at 479.5 Mb, was positioned near marker *AX-111708401* (473.9 Mb), identified by Wang et al. [53]. According to Bhatta et al. [39], a locus with a negative effect towards GZnC was found on the long arm of chromosome 4A, which is consistent in location and effect with the QTL in our study (Appendix A). Recently, Ma et al. [52], analyzing a Chinese diversity panel of spring bread wheat, detected MTAs for GFeC in chromosome 7A in the region 462.33 Mb, which was colocated with locus *QFe.icg-7A* determined in our study (Appendix A). Published data indicate that most loci that coincide in chromosomal localization belong to the genome of common wheat.

### 3.2. Wheat Relatives as Sources of Potentially Valuable Loci for Increased Zinc and Iron Content in Grain

Wild and cultivated relatives of wheat demonstrate a much wider range of iron and zinc concentrations in their grains compared to elite cultivars of common wheat [20,55,56]. These species are therefore being considered as additional sources for improving GFeC and GZnC in cultivated wheat varieties. For example, a GWAS on *Aegilops tauschii*, a wild progenitor of the D genome for common wheat, revealed MTAs associated with micronutrient concentrations, including Fe and Zn, in grain [50]. Herewith, five MTAs with GFeC were detected on chromosomes 1D, 2D, 3D, 4D, and 7D, while four SNPs associated with GZnC were found on chromosomes 2D, 4D, 6D, and 7D. A number of QTLs for GZnC and GFeC, characteristic of *T. durum*, *T. dicoccum*, and *T. dicoccoides*, were found on chromosomes 1B, 2A, 2B, 4A, 4B 5A, 5B, 7A, and 7B using QTL mapping and GWASs [41,57,58,59]. According to our study, chromosomes 1B, 2A, 2B, 5A, 5B, and 5D of introgression lines with *T. dicoccum* genetic material also contain loci that have a positive effect on zinc and iron content in grain (Table 1). However, the localization of these loci does not coincide with previously published data.

In our study, we focused on MTAs derived from wheat relatives with other genomes: *T. timopheevii* (A^t^A^t^GG) and *T. kiharae* (A^t^A^t^GGDD). Analysis of MTAs specific to *T. timopheevii*, *T. kiharae*, and ILs containing genome fragments from these species revealed four loci on chromosomes 1B (two loci), 2A, and 2B and seven single SNPs inherited from wheat relatives. It should be noted that most alleles from wheat relatives mapped in this study were associated with increased Fe and Zn content, except SNPs from locus *QZn.icg-1B.2*, as well as *Tdurum_contig64416_441* (chromosome 2B) and *Excalibur_c82693_359* (chromosome 5B), which negatively affected Zn content (Table 1). Additionally, two SNPs on loci *QZnFe.icg-2A* and *QZnFe.icg-2B* were associated with the contents of both elements.

The data on the contribution of genotype and environment to the variability of GZnC and GFeC traits are ambiguous. Several studies have shown that soil and climatic conditions play a major role in the variation in zinc and iron content. Shamanin et al. [60], in their analysis of the zinc content in the grains of spring wheat varieties, found that the primary contribution to the trait variability came from the ‘environments’ factor (38.7%), which was influenced by differences in soil Zn content and moisture availability. Another study indicated that Fe and Zn levels in spring and winter common wheat genotypes were largely controlled by location [61]. Interesting data were obtained by Peleg et al., who analyzed GZnC and GFeC in wild emmer wheat under contrasting irrigation regimes [22]. Their findings showed that wild emmer accessions exhibited higher genetic diversity for these traits, with stronger genotype and genotype-by-environment interaction effects under both drought and irrigated conditions, compared to durum cultivars. In our study, ANOVA revealed that GZnC and GFeC were significantly influenced by genotype, environment, and genotype-by-environment interaction effects, with genotype contributing the most to trait variation, followed by genotype x environment interactions (Appendix A). Given the moderately high broad-sense heritability of both traits (>0.67), we can assume a higher effect of wild loci on the phenotypic expression of these traits under favorable environmental conditions.

At present, there is no published information on mapping QTLs for GZnC and GFeC derived from species with genomes A^t^A^t^GG. This allows us to conclude that we have obtained new data indicating the presence of effective QTLs specific to the species of the Timopheevi group. It is important to note that the *QZnFe.icg-2A* locus detected in this study is located near a previously identified locus inherited from *T. timopheevii* and associated with increased gluten content [45]. Positive correlations between some trace element concentrations (Zn, Fe) and gluten contents allows for the simultaneous improvement of several quality parameters. Three important MTAs from *QZnFe.icg-2A* loci and four MTAs from *QZnFe.icg-2B* (Table 1) may be used for the development of KASP markers to screen breeding germplasm in the further study.

### 3.3. Candidate Genes

The selected candidate genes were annotated as iron or zinc transporters, representing diverse gene families. The regulation of metal ion content in cereal grain involves intricate mechanisms encompassing uptake, transport, and storage [62,63]. Upon absorption by root cells, iron (Fe) and zinc (Zn) are radially transported into the xylem through the root epidermis, cortex, and endodermis. Subsequently, they are unloaded into the phloem for transport to their final long-distance destinations. Various families of transporter genes play pivotal roles in the transportation of Fe and Zn in wheat, with the metal ion transport system being a critical aspect of plant growth and development [64,65].

Three genes (*TraesCS1A03G0312000* and *TraesCS1A03G0341000* from the *QZn.icg-1A.1* locus and *TraesCS7A03G0836800* from *QFe.icg-7A*) were annotated as ZIP zinc/iron transporters (Table 2). Previously, ZIP transporters were shown to be involved in Zn absorption from the soil, with subsequent remobilization and accumulation in the grain, as identified in rice [66]. For example, the genes *OsZIP1*, *OsZIP3*, *OsZIP4*, and *OsZIP5* have been reported to be overexpressed during Zn deficiency [67,68]. Within *QZn.icg-1A.1*, *TraesCS1A03G0312000* (*TraesCS1A02G125500*) is homologous to the rice gene *OsZIP7* (*Os05g0198400*), which encodes a zinc/iron transporter. Yang et al. [69] demonstrated that *OsZIP7* is strongly expressed in rice roots under Fe-deficient conditions and can complement an Fe uptake-deficient yeast mutant. Furthermore, Kumari et al. [62] suggested that *OsZIP7* may be involved in zinc uptake from the rhizosphere. *TraesCS1A03G0312000* is most strongly expressed in the flag leaf during the senescence stage, suggesting its role in Zn retransportation from leaves to grain during plant aging, though its involvement in metal ion uptake from the environments seems unlikely. Meanwhile, *TraesCS1A03G0341000*, from the same locus on chromosome 1A, shows pronounced expression in roots, flag leaves, and grain in wheat plants. *TraesCS7A03G0836800* is most strongly expressed in roots throughout the vegetative and early reproductive stages of plant development. These expression patterns suggest that their involvement in metal ion accumulation in the grain is likely related to transport, including uptake from the soil via the roots and subsequent translocation.

*TraesCS4A03G0233800* encodes a mitochondrial iron transporter (MIT), which is involved in transporting iron into mitochondria. MIT has been shown to play a significant role in mitochondrial iron uptake and distribution in plants, contributing to their growth [70]. While this gene does not have a clearly defined expression pattern, it is most strongly expressed in grain during the ripening stage.

On the *QFe.icg-5B.1* locus, a gene encoding metal ion transporters was identified. *TraesCS5B03G0236100* encodes Ferroportin (FPN), an iron efflux transporter. In *Arabidopsis*, Ferroportins are plasma membrane-localized exporters of Fe and Co, expressed in the vasculature of the root and shoot, and are involved in loading these elements from the pericycle into the xylem [71]. Analysis of FPN homologs in rice demonstrated that *OsFPN1* can transport other metal ions, such as Co and Ni, and is essential for detoxification in rice [72]. *TraesCS5B03G0236100* was strongly expressed in the fifth and flag leaf blades, especially at night during the flag leaf stage, which may indicate the possible diurnal regulation of metal ion transportation processes.

Three genes within the *QFe.icg-2B.2* locus—*TraesCS2B03G1038000*, *TraesCS2B03G1038100*, and *TraesCS2B03G1040500*—belong to the same gene family, encoding metal-nicotinamide transporter YSL-like proteins. One of these genes, *TraesCS2B03G1038000*, previously annotated as *TaYSL9-2B*, is proposed to function as a Fe^+2^-NA and Fe^+3^-DMA transporter [63,73]. The expression of this gene was significantly increased under iron-deficient conditions. In rice, *OsYSL9* is involved in exporting iron from the endosperm to the embryo [74]. The increased expression of *TaYSL9* in the grains of iron-deficient plants suggests a potential parallel function with *OsYSL9* [73]. The expression patterns of the identified *YSL*-like genes were similar to those of *TaYSL9* (Appendix A). All three genes were expressed in most tissues during plant development, including roots, leaves, and spikes, with predominant expression in flag leaves after ear emergence. This may indicate that *YSL*-like genes are involved in iron remobilization from the flag leaves to the grains, a mechanism previously proposed by Wang et al. for *TaYSL9* [73].

In this study, we identified the candidate genes *TraesCS1A03G0312000*, *TraesCS1A03G0341000*, *TraesCS2B03G1038000*, *TraesCS2B03G1038100*, *TraesCS2B03G1040500*, and *TraesCS5B03G0236100*, which may complement one of the potential pathways of metal ion concentration in wheat grain - iron and zinc translocation from flag leaves to seeds. Other genes may be involved in metal ion transport in different tissues: *TraesCS7A03G0836800*, primarily expressed in roots, represents the mechanism by which metal ions are taken up from the soil, followed by their transportation upstream in the plant, while *TraesCS4A03G0233800*, expressed in grains, indicates a pathway for their accumulation in this tissue.

## 4. Materials and Methods

### 4.1. Plant Material Phenotyping

A collection of 157 spring bread wheat (*T. aestivum* L.) samples was used in this study, including modern Russian varieties, landraces, and introgression lines (ILs) obtained from the hybridization of bread wheat cultivars with tetraploid relatives *T. durum* (genome AABB), *T. dicoccum* (AABB), *T. dicoccoides* (AABB), and *T. timopheevii* (A^t^A^t^GG) and a synthetic hexaploid line, *T. kiharae* (A^t^A^t^GGDD). A list of plant materials is presented in Appendix A. Information on the chromosomal localization of foreign genetic material, obtained using genotyping with SSR and SNP markers, can be found in [45].

The plant material was grown in duplicate with a randomized block design in the experimental field of the Institute of Cytology and Genetics SB RAS (Novosibirsk region, 54 91’91″ N, 82 99’03″ E) in 2018–2019. The varieties were sown between 15 and 20 May in both years. The trials were harvested in the first week of September. The growing season of 2018 was characterized by low temperatures in May (on average, 5 °C below normal) and high waterlogging in May–June. The weather conditions in 2019 were characterized by high temperature in the second half of the growing season. Rainy weather was observed in May of 2019, and a drought was observed in July and August. Overall, the 2019 growing season was significantly warmer than the 2018 (by 4.6 °C above normal). Samples were sown in plots 1 m wide, with 60–80 grains per row and a distance of 25 cm between rows. Standard agronomic procedures were used for field cultivation. The soil of the experimental field consisted of leached chernozem; the fertile soil layer varied within 40–60 cm, and the humus content was 4.2%, with pH 6.7. The contents of nitrogen, phosphorus, and potassium were 0.34%, 0.30%, and 0.13%, respectively; the gross content of zinc was 32.5 mg/kg. After harvesting, the grains were air-dried, and 3 g of grain was selected for analysis. The concentration of zinc and iron was assessed using an atomic absorption spectrometer ContrAA 800 D with flame atomization (Analytik Jena, Jena, Germany) according to the manufacturer’s protocol.

### 4.2. Statistical Analysis

Descriptive statistics and analysis of variance (ANOVA) were calculated using R and the Statistica v. 10 software package. The best linear unbiased estimates (BLUEs) for each accession were used, assuming the genotype as a fixed effect and the growing season as a random effect. The significance of differences between the mean values of the two sample sets was determined using Student’s *t*-test. Spearman’s correlation coefficients (r) were calculated to explore the association between GFeC and GZnC. Heritability estimates were calculated using the formula H^2^ = σ_g_/(σ_g_ + σ_e_/n), where σ_g_ is a genotypic variance, σ_e_ is an error variance, and n is the number of environments.

### 4.3. Genotyping and Genome-Wide Association Study

DNA was extracted following the modified sodium bisulfite protocol described in [75]. DNA purification for SNP genotyping was performed using a Bio-Silica Kit for DNA Purification from Reaction Mixtures according to the manufacturer’s protocol. The DNA was then quantified using the Qubit dsDNA BR Assay kits (Thermo Fisher Scientific, Waltham, MA, USA) on a Qubit 4 Fluorometer (Thermo Fisher Scientific). SNP genotyping was performed using the Illumina Infinium 15K Wheat platform by TraitGenetics-Section of SGS Institute Fresenius GmbH (https://www.traitgenetics.com (accessed on 22 August 2024)). The array comprised 13,006 SNP markers. Markers with a minor allele frequency (MAF) of less than 10% and missing data exceeding 10% were excluded from the genotype dataset.

The genotyping data included 10,611 filtered SNPs. Phenotypic data for each year, phenotypic data normalized by the logarithm of 10, and Best Linear Unbiased Estimates (BLUEs) accounting for both years were utilized for whole-genome association analysis using the R package GAPIT [76]. Narrow-sense heritability for each trait was estimated using the MLM.

For each trait, a mixed linear model (MLM) considering kinship and population structure (principal components), Fixed and Random Model Circulating Probability Unification (FarmCPU), and Bayesian Information and Linkage Disequilibrium Iteratively Nested Keyway (BLINK) were applied. The best model was selected based on the Q-Q plots. Quantile-quantile (Q-Q) and Manhattan plots were generated using the R package CMplot [77]. Q-Q plots are presented in Appendix A.

The chromosomal locations of the SNPs were determined using the International Wheat Genome Sequencing Consortium (IWGSC) RefSeq v.1.1 and RefSeq v.2.1, as submitted in the GrainGenes genomic browser [46] and the consensus genetic linkage map [47].

### 4.4. Gene Annotation and Prioritization

We conducted gene prioritization for quantitative trait loci (QTLs) identified by markers showing significant associations with the analyzed traits. The boundaries of these loci were determined based on markers separated by a linkage disequilibrium (LD) distance, as illustrated in Appendix A. LD estimates were obtained using the Genetics R package [78], and LD decay plots were generated using the R package LDheatmap [79].

The list of genes and their functional annotations was retrieved from the IWGSC RefSeq v.2.1 deposited in the genome browser Persephone [80]. Genes were annotated based on Gene Ontology terms using AgriGo v2 [81]. Candidates were selected based on their correspondence to the following GO terms: GO:0009987 (cellular response to iron ion starvation), GO:0034605 (iron ion transport), GO:0046348 (zinc II ion transport), GO:0051656 (zinc II ion transmembrane transport), GO:0051651 (iron–sulfur cluster assembly), GO:0051495 (iron ion homeostasis), GO:0051493 (response to zinc ion), GO:0045851 (response to iron ion), GO:0006281 (2 iron, 2 sulfur cluster binding), GO:0015849 (iron ion binding), GO:0045859 (iron–sulfur cluster binding), GO:0050832 (zinc ion transmembrane transporter activity). Next, we compared the resulting selection with a list of genes obtained from KnetMiner [48] using the keywords “iron and grain”, “zinc and grain”, “iron concentration”, and “zinc concentration”. Additionally, we examined functional annotations for compliance with protein families known to be involved in iron/zinc content. Genes associated with GZnC and GFeC have been shown to be expressed in roots, shoots, leaves, flowers, and grain tissues [51,73]. Therefore, it is not possible to identify a single or typical expression pattern for genes involved in metal ion content in grain. As a result, we did not use expression patterns as a parameter for gene prioritization. The expression patterns of the selected candidate genes were assessed using the developmental time course data of the common wheat cultivar Azhurnaya [82] available in Wheat Expression Browser expVIP [83].

## 5. Conclusions

Our study discovered new genetic loci from species of the Timopheevi group with A^t^A^t^GG genome controlling Zn and Fe accumulation in wheat grain. Importantly, our data revealed a colocalization of QTLs for grain Zn, grain Fe content, and gluten content that suggests the possibility of simultaneous breeding for these traits. The MTAs identified herein may help to develop suitable KASP markers for the validation of desirable loci and use them for improvement zinc and iron content in wheat varieties via marker-assisted selection. Efforts will also be aimed at assessing the impact that the new locus has on the yield and quality traits of bread wheat. Introgression lines showing high stability over various environments should be carefully selected as donor parents for breeding programs.

## Figures and Tables

**Figure 1 ijms-25-10556-f001:**
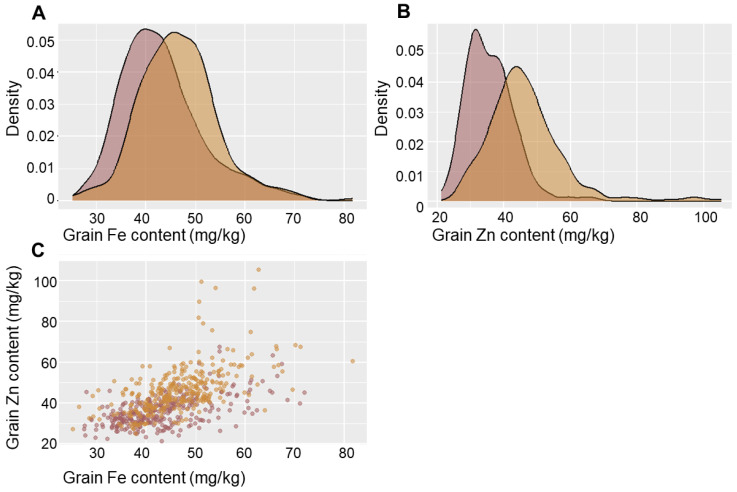
Probability density for (**A**) grain iron content and (**B**) grain zinc content, and (**C**) linear regressions for Zn vs. Fe in 2018 (red) and 2019 (orange) in a set of 157 common wheat varieties.

**Figure 2 ijms-25-10556-f002:**
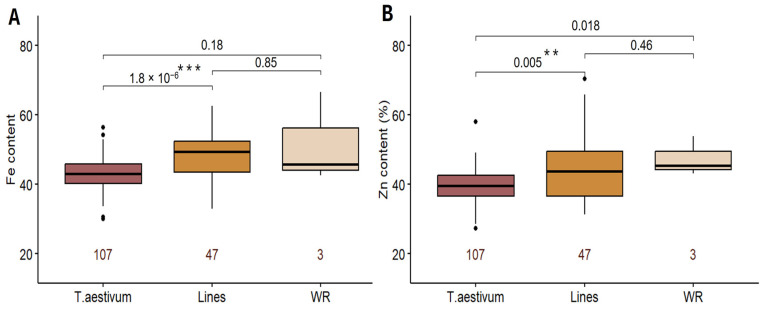
Box plots showing the variation in (**A**) Fe and (**B**) Zn content among spring wheat varieties (*T. aestivum*), introgression lines (Lines), and wheat relatives (WRs) across two environments (2018–2019). The number of genotypes in each group is indicated below the bars. *p*-values are indicated above the bars (** *p* < 0.01, *** *p* < 0.001). Black dots denote outliers.

**Figure 3 ijms-25-10556-f003:**
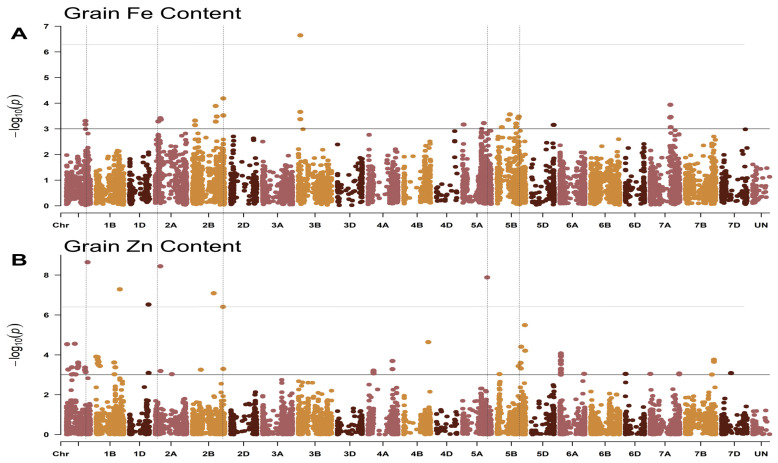
Manhattan plots of genome-wide association study (GWAS) results obtained using the BLINK model for (**A**) Fe and (**B**) Zn content in common wheat varieties, introgression lines, and wheat relatives. The horizontal black line indicates the threshold of significance *p* < 0.001, and the upper horizontal gray line indicates the threshold of significance with *p*-value < 0.05. The vertical dotted lines indicate the significant markers common to both GFeC and GZnC.

**Table 1 ijms-25-10556-t001:** Significant QTLs and single SNP markers associated with Zn and Fe concentration and inherited from *T. timopheevii*, *T. kiharae* and *T. dicoccum*.

SNP	Chr	QTL	Position RefSeq 1.1, bp	Position RefSeq 2.1, bp	Genetic Position, cM	*p*-Value	Allele	Effect	Allele Source
*BS00025736_51*	1B	*QZn.icg-1B.1*	39,866,100	44,233,164	56.65	2.7 × 10^−4^	G	Zn + 4.95	*T.timopheevii*, *T. kiharae*, *T.dicoccum*
*Kukri_rep_c100936_449*			40,199,636	44,565,803	56.88	1.8 × 10^−4^	T	Zn + 5.29	*T. timopheevii*, *T. kiharae*, *T.dicoccum*
*GENE-0427_442*			40,942,799	45,304,682	56.88	1.3 × 10^−4^	A	Zn + 5.45	*T. timopheevii*, *T. kiharae*, *T.dicoccum*
*Excalibur_c17202_1833 RAC875_c826_839*	1B	*QZn.icg-1B.2*	58,324,982 59,599,775	63,893,763 65,164,043	57.59 57.59	3.2 × 10^−4^ 2.2 × 10^−4^	G G	Zn − 2.76 Zn − 2.89	*T. timopheevii*, *T. kiharae* *T. timopheevii*, *T. kiharae*
*Kukri_c44442_274*	2A	*QZn.icg-2A*	123,548,354	128,154,513	102.43	3.6 × 10^−9^	C	Zn + 4.05	*T. timopheevii*, *T. kiharae*, *T. dicoccum*
*Kukri_c441_891*			126,655,033	131,241,681	102.47	3.8 × 10^−4^	G	Zn + 5.07 Fe + 3.27	*T. timopheevii*, *T. kiharae*
*TA001792-1026*			141,166,705	145,801,217	103.62	4.3 × 10^−4^	G	Fe + 3.63	*T. timopheevii*, *T. kiharae*
*CAP12_s9114_61*	2B	*QZn.icg-2B*	767,132,274	776,298,229	134.46	3 × 10^−4^	T	Fe + 4.61	*T. timopheevii*, *T. kiharae*
*BobWhite_c5191_362*			767,169,885	776,335,465	134.46	3 × 10^−4^	T	Fe + 4.60	*T. timopheevii*, *T. kiharae*
*BobWhite_c5191_563*			767,170,156	776,336,169	134.46	6.6 × 10^−5^	T	Zn + 2.61 Fe + 5.14	*T. timopheevii*, *T. kiharae*
*BS00046601_51*			767,375,331	776,541,318	134.46	2.9 × 10^−4^	C	Fe + 4.52	*T. timopheevii*, *T. kiharae*
*Excalibur_c14859_394*	1B		588,766,841	595,748,769	1A: 102.92	5.2 × 10^−8^	G	Zn + 4.58	*T. timopheevii*, *T. kiharae*
*Ku_c6546_718*	2B		206,752,135	214,819,365	97.54	5.6 × 10^−4^	T	Zn + 3.21	*T. timopheevii*, *T. kiharae*, *T. dicoccum*
*Tdurum_contig64416_441*	2B		530,046,526	538,048,972	102.23	8.2 × 10^−8^	A	Zn − 2.07	*T. timopheevii*, *T. kiharae*
*BS00067150_51*	5A		609,243,940	611,013,091	98.72	2.1 × 10^−7^	A	Zn + 4.08	*T. timopheevii*, *T. kiharae*, *T. dicoccum*
*Excalibur_c82693_359*	5B		604,022,712	607,850,798	unmap	3.9 × 10^−5^	T	Zn − 3.83	*T. timopheevii*, *T. kiharae*
*BS00091302_51*	5B		695,491,227	7B:67,683,834	58.88	3.2 × 10^−6^	C	Zn + 4.28	*T. timopheevii*, *T. kiharae*, *T. dicoccum*
*RAC875_c703_2551*	5D		555,048,619	558,933,322	202.99	7 × 10^−4^	T	Fe + 5.18	*T. timopheevii*, *T. kiharae*, *T. dicoccum*

Chr, chromosome. Regarding QTL, the name of the loci is assigned to the QIL that includes two or more SNPs; SNP positions are given according to RefSeq v.1.1 and RefSeq v. 2.1 [46]; and genetic positions (cM) are based on Wang et al. [47].

**Table 2 ijms-25-10556-t002:** Candidate genes for grain zinc and iron content.

QTL	Chr	Gene ID IWGSC RefSeq v2.2	Gene ID IWGSC RefSeq v1.1	Description
*QZn.icg-1A.1*	Chr1A	*TraesCS1A03G0312000*	*TraesCS1A02G125500*	Zinc/iron permease
*QZn.icg-1A.1*	Chr1A	*TraesCS1A03G0341000*	*TraesCS1A02G133400*	ZIP metal ion transporter family
*QFe.icg-2B.2*	Chr2B	*TraesCS2B03G1038000*	*TraesCS2B02G407900*	Metal-Nicotianamine Transporter YSL2
*QFe.icg-2B.2*	Chr2B	*TraesCS2B03G1038100*	*TraesCS2B02G408000*	Metal-Nicotianamine Transporter YSL2
*QFe.icg-2B.2*	Chr2B	*TraesCS2B03G1040500*	*TraesCS2B02G408500*	Metal-Nicotianamine Transporter YSL2
*QZn.icg-4A.1*	Chr4A	*TraesCS4A03G0233800*	*TraesCS4A02G113800*	Mitochondrial iron transporter
*QFe.icg-5B.1*	Chr5B	*TraesCS5B03G0236100*	*TraesCS5B02G093600*	Ferroportin1 (FPN1), iron-regulated protein 3
*QFe.icg-7A*	Chr7A	*TraesCS7A03G0836800*	*TraesCS7A02G340000*	ZIP Zinc transporter

## Data Availability

All the data presented in this study are included in the manuscript and Appendix A.

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
