# Peer review of "Identification of Genomic Regions Conferring Enhanced Zn and Fe Concentration in Wheat Varieties and Introgression Lines Derived from Wild Relatives"

_ijms, 2024, doi:10.3390/ijms251910556_

Round 1

Reviewer 1 Report

Comments and Suggestions for Authors

This study used a wheat panel consisting 153 varieties and lines to identify the marker-trait association (MTAs) loci for GZnC and GFeC by GWAS, and some candidate genes were predicted.    These results will provide some important information for improving the grain micronutrients Zn and Fe content in future breeding. But it still needs to be improved in this paper.

1. It is well known that the soil minerals content will affect the minerals content of wheat grain, so in Materials and Methods, the soil different minerals content should be introduced. In addition, to conduct GWAS analysis, a minimum of 3 environments are required, but in this paper there is only two environments.  

2. In Figure1, different photos should be noted A, B, C, D so that it will be clear. So do other figures.

3. In Figure3, important MAT loci should be marked using different bright colors.

4. The authors used the IWGSC RefSeq v.2.1 to predict the candidate genes, but in table2,  IWGSC RefSeq v.1.1 was found, why? Some candidate genes for one locus are different.

5. The format of references was inconsistent, such as line480. And the number of references should be reduced.

6. The languages should be further improved.

Comments on the Quality of English Language

The languages should be improved.

Author Response

Dear Reviewer, 
The authors are grateful to Reviewer for a careful reading of the manuscript and valuable comments. In general, we agree with all the reviewers' suggestions for improving the manuscript. 
Below are answers to the comments:

 Reviewer:

This study used a wheat panel consisting 153 varieties and lines to identify the marker-trait association (MTAs) loci for GZnC and GFeC by GWAS, and some candidate genes were predicted. These results will provide some important information for improving the grain micronutrients Zn and Fe content in future breeding. But it still needs to be improved in this paper.

Comment 1. It is well known that the soil minerals content will affect the minerals content of wheat grain, so in Materials and Methods, the soil different minerals content should be introduced. In addition, to conduct GWAS analysis, a minimum of 3 environments are required, but in this paper there is only two environments.

Response 1: Information about soil and weather condition was added in Material and Methods (lines 420-431). We completely agree with Reviewer that reliability of GWAS analysis and marker-trait association depends on the number of experiments conducted. According to the literature, GWAS requires at least two years of trait evaluation, or trait in two different locations with different environmental conditions. A number of studies have shown that two-year experiments are sufficient to obtain significant marker-trait associations. For example:

  • Shepelev et al. Variation of Macro- and Microelements, and Trace Metals in Spring Wheat Genetic Resources in Siberia. Plants 2022, 11, 149. https://doi.org/10.3390/plants11020149
  • Bhatta et al. Genome-Wide Association Study Reveals Novel Genomic Regions Associated with 10 Grain Minerals in Synthetic Hexaploid Wheat. Int. J. Mol. Sci. 2018, 19, 3237; doi:10.3390/ijms19103237
  • Kumar et al. Euphytica (2018) 214:219 https://doi.org/10.1007/s10681-018-2284-2
  • Gerard et al. Genome-wide association mapping of genetic factors controlling Septoria tritici blotch resistance and their associations with plant height and heading date in wheat. Euphytica 213, 27 (2017). https://doi.org/10.1007/s10681-016-1820-1
  • Hao et al. Genome-wide association study of grain micronutrient concentrations in bread wheat. Journal of Integrative Agriculture 2024, 23, 1468-1480 https://doi.org/10.1016/j.jia.2023.06.030

Comment 2. In Figure1, different photos should be noted “A, B, C, D” so that it will be clear. So do other figures.  

Response 2: Figure 1 was corrected.

Comment 3. In Figure3, important MAT loci should be marked using different bright colors. 

Response 3: Figure 3 was corrected.

Comment 4. The authors used the IWGSC RefSeq v.2.1 to predict the candidate genes, but in table2, IWGSC RefSeq v.1.1 was found, why? Some candidate genes for one locus are different.  

Response 4: For comparison of the results in Table 2 and Table 3, we added column with RefSeq v.2.1 data.
A physical map of the pseudomolecule (RefSeq v.1.1 and v. 2.1) was used to identify candidate genes. However, the exact position of candidate genes may differ in the experimental population. That's why we expanded the region on the chromosome for candidate gene analysis by 100 – 250 kb to capture genes only closely linked to the significant markers which passed the FDR threshold and located within the LD interval. The annotated genes within 250 kb of the mapped SNP were considered candidate genes as described in other papers (see Ahmed et al. Agriculture 2020, 10, 392; doi:10.3390/agriculture10090392). That's why some candidate genes were included in the QTLs.

Comment 5. The format of references was inconsistent, such as line480. And the number of references should be reduced. 

Response 5: Reference [14] was corrected. A chapter from this book has been provided (line 549-551). We think that our article contains the required number of references. But we are ready to reduce them according to the editor's decision.

Comment 6. The languages should be further improved.

 Response 6 : language correction was done

With kind regards
Authors

Reviewer 2 Report

Comments and Suggestions for Authors

1.     Given the availability of a vast amount of genetic diversity of wheat in terms of cultivated and wild wheat, the study has focused on the identification of the loci associated with grain zinc and grain iron and also the genomic regions linked to these traits. The authors identified various marker trait association (MTA) for grain iron and grain zinc out of which this study found 19 novel MTAs. Overall, the study provides valuable insights for breeding new wheat varieties with higher iron and zinc content.

2.     Line 26: Replace “one a significant” with “one of the significant”.

3.     Line 367-370: Please specify the months during which this field experiment was conducted to understand the environmental conditions.

4.     Line 372: IS there any standard protocol to measure the iron and zinc content or any literature available, if so, please briefly mention or provide citation.

5.     Line 164: Please italicized T. aestivum

6.     Please discuss what could be the important factors contributing to higher the metal contents in 2019 (Figure 1 results).

7.     Is there any gene expression data associated with the selected candidate genes which were annotated? It would be great to add that data, if available.

8. Please proofread the manuscript carefully. 

Author Response

Dear Reviewer,

The authors are grateful to reviewer for a careful reading of the manuscript, valuable comments and English correction. In general, we agree with all the reviewers' suggestions for improving the manuscript.

Below are answers to the comments:

 Reviewer

Comment 1. Given the availability of a vast amount of genetic diversity of wheat in terms of cultivated and wild wheat, the study has focused on the identification of the loci associated with grain zinc and grain iron and also the genomic regions linked to these traits. The authors identified various marker trait association (MTA) for grain iron and grain zinc out of which this study found 19 novel MTAs. Overall, the study provides valuable insights for breeding new wheat varieties with higher iron and zinc content.

Response 1: Thank you very much for your kind opinion about our manuscript

Comment 2. Line 26: Replace “one a significant” with “one of the significant”.

 Response 2: corrected.

Comment 3. Line 367-370: Please specify the months during which this field experiment was conducted to understand the environmental conditions.

Response 3: Information about soil and weather condition was added in Material and Methods (lines 420-431).

 Comment 4. Line 372: IS there any standard protocol to measure the iron and zinc content or any literature available, if so, please briefly mention or provide citation.

Response 4: Currently, various mass spectrometers that do not require labor-intensive sample preparation are used to analyze micro- and macroelements in cereal grain. For example, an atomic absorption spectrophotometer (AAS) (https://doi.org/10.3390/genes13061052), plasma mass spectrometer ICP-MS (https://doi.org/10.3390/ijms19103237). With the help of such equipment, it is possible to conduct direct analysis of solid forms (grains, powders, fibers). The protocols of the manufacturers of this equipment are standardized and allow to compare the results obtained in different studies. In our work, atomic absorption spectrometry with flame atomization (ContrAA 800 D, Analytik Jena, Germany) was used.

Comment 5. Line 164: Please italicized T. aestivum

 Response 5: corrected.

Comment 6. Please discuss what could be the important factors contributing to higher the metal contents in 2019 (Figure 1 results). 

Response 6: Short discussion was added (lines 316-332)

Comment 7. Is there any gene expression data associated with the selected candidate genes which were annotated? It would be great to add that data, if available.

Response 7: we try to analyze gene expression data. We add new information in section 3.3 of Discussion (line 354-408) and prepare Table S7 with expression patterns of candidate genes.

Comment 8. Please proofread the manuscript carefully.

 Response 8: We try to check manuscript carefully

With kind regards
Authors

Reviewer 3 Report

Comments and Suggestions for Authors

The manuscript by Leonova et al. entitled “Identification of genomic regions conferring enhanced Zn and Fe concentration in wheat varieties and introgression lines derived from wild relatives” report the mineral Fe and Zn contents in grains of wheat varieties and introgression lines. The SNPs among the varieties are user to mine the mineral contents related genes. The work is technically sound piece of research, and within the scope of IJMS, however, some suggestions maybe fixed before acceptance.

1.       Mining the expression profiles of the candidate genes

2.       Predict HUB genes and the pathway

Author Response

Dear Reviewer,

The authors are grateful to reviewer for a careful reading of the manuscript, valuable comments and English correction. In general, we agree with all the reviewers' suggestions for improving the manuscript.

Below are answers to the comments:

 Reviewer 3:

The manuscript by Leonova et al. entitled “Identification of genomic regions conferring enhanced Zn and Fe concentration in wheat varieties and introgression lines derived from wild relatives” report the mineral Fe and Zn contents in grains of wheat varieties and introgression lines. The SNPs among the varieties are user to mine the mineral contents related genes. The work is technically sound piece of research, and within the scope of IJMS, however, some suggestions maybe fixed before acceptance.

 Responce: Thank you very much for your kind opinion about our manuscript

Comment 1. Mining the expression profiles of the candidate genes

 Response 1: we add Table S7 with expression profiles of the candidate genes and add comments in section 3.3 (line 354-408).

Comment 2. Predict HUB genes and the pathway

Response 2: we expanded section 3.3 in the Discussion chapter (line 354-408).

With kind regards
Authors
